# Antiviral Activity of *Ficus rubiginosa* Leaf Extracts against HSV-1, HCoV-229E and PV-1

**DOI:** 10.3390/v14102257

**Published:** 2022-10-14

**Authors:** Federica Dell’Annunziata, Carmine Sellitto, Gianluigi Franci, Maria Carla Marcotullio, Anna Piovan, Roberta Della Marca, Veronica Folliero, Massimiliano Galdiero, Amelia Filippelli, Valeria Conti, Domenico Vittorio Delfino

**Affiliations:** 1Department of Experimental Medicine, University of Campania “Luigi Vanvitelli”, 80138 Naples, Italy; 2Department of Medicine Surgery and Dentistry, University of Salerno, Baronissi, 84081 Salerno, Italy; 3Clinical Pharmacology Unit, San Giovanni di Dio e Ruggi D’Aragona University Hospital, 84131 Salerno, Italy; 4Clinical Pathology and Microbiology Unit, San Giovanni di Dio e Ruggi D’Aragona University Hospital, 84131 Salerno, Italy; 5Department of Pharmaceutical Sciences, University of Perugia, Via del Liceo 1, 06122 Perugia, Italy; 6Department of Pharmaceutical and Pharmacological Sciences, University of Padova, Via F. Marzolo 5, 35100 Padova, Italy; 7Department of Medicine, Section of Pharmacology, University of Perugia, 06122 Perugia, Italy

**Keywords:** *Ficus rubiginosa*, HSV-1, HCoV-229E, PV-1, antiviral activity, leaf extract, natural compound

## Abstract

*Ficus rubiginosa* plant extract showed antimicrobial activity, but no evidence concerning its antiviral properties was reported. The antiviral activity of the methanolic extract (MeOH) and its *n*-hexane (H) and ethyl acetate (EA) fractions against *Herpes simplex* virus-1 (HSV-1), Human coronavirus (HCoV) -229E, and Poliovirus-1 (PV-1) was investigated in the different phases of viral infection in the VERO CCL-81 cell line. To confirm the antiviral efficacy, a qPCR was conducted. The recorded cytotoxic concentration 50% was 513.1, 298.6, and 56.45 µg/mL for MeOH, H, and EA, respectively, assessed by 3-(4,5-dimethylthiazol-2-yl)-2,5-diphenyl-2H-tetrazolium bromide (MTT) assay after 72 h of treatment. The *Ficus rubiginosa* leaf extract inhibited the replication of HSV-1 in the early stages of infection, showing a complete inhibition up to 0.62, 0.31, and 1.25 µg/mL. Against HCoV-229E, a total inhibition up to 1.25 µg/mL for MeOH and H as well as 5 µg/mL for EA was observed. Otherwise, no activity was recorded against PV-1. The leaf extract could act directly on the viral envelope, destructuring the lipid membrane and/or directly blocking the enriched proteins on the viral surface. The verified gene inhibition suggested that the treatments with M, H, and EA impaired HSV-1 and HCoV-229E replication, with a greater antiviral efficiency against HSV-1 compared to HCoV-229E, possibly due to a greater affinity of *Ficus rubiginosa* towards membrane glycoproteins and/or the different lipid envelopes.

## 1. Introduction

Viral infections represent a serious global health problem with significant morbidity and direct/indirect social and economic impacts [1]. Several outbreaks of emerging infectious diseases have threatened health security and the world economy [2,3,4,5]. Furthermore, the COVID-19 pandemic caused by the severe acute respiratory syndrome coronavirus 2 (SARS-CoV-2) highlighted the need to discover broad-spectrum therapies that could be used to counter future viral threats [6].

Besides the emerging infections, a major public health concern is the development of antimicrobial resistance due to drug overuse [7]. It is estimated that 700,000 annual deaths are attributed to multidrug-resistant pathogens [8]. Contextually, a herpes simplex virus (HSV) infection represents one of the most common infectious diseases in the population [9]. HSV can be classified into two types: HSV type 1 (HSV-1), mainly transmitted by oral-to-oral contact and causing the common herpes labialis, but which can also be associated with genital herpes [10], and HSV type 2 (HSV-2), the causative agent of genital herpes, which is sexually transmitted [11]. The World Health Organization (WHO) estimated that approximately 3.7 billion and 491 million people aged < 50 are affected by HSV-1 and HSV-2, respectively [9]. To date, the common drugs for HSV treatment are acyclovir, valacyclovir, and penciclovir [12]. However, the emergence of drug resistance and the inability of these drugs to eradicate latent infections underline the need for new exploratory strategies [13]. Another serious threat to public health is represented by the *Coronaviridae* family. Seven coronaviruses are known to infect humans [14]. Human coronavirus (HCoV) -229E, -NL63, -OC43, and -HKU1 are associated with common cold symptoms, while severe acute respiratory syndrome coronavirus (SARS-CoV), Middle Eastern respiratory syndrome (MERS-CoV), and SARS-CoV-2 are highly pathogenic [15,16]. However, although HCoV-229E is associated with mild respiratory symptoms, its presence can cause serious complications, especially in conditions of co-infections with other microorganisms [17]. Another important pathogen is poliovirus (PV-1), the causative agent of poliomyelitis. It belongs to the family Picornaviridae, of the genus Enterovirus. Its infection occurs via fecal–oral transmission, through the ingestion of contaminated water or food, or through droplets of saliva [18]. Although polio has been eradicated in several countries, there are numerous outbreaks in developing countries each year [19].

The search for new therapies to combat emerging and multi-drug-resistant infections finds encouraging approaches in ethnopharmacology [20]. The analysis of plant extracts revealed the presence of bioactive metabolites, which act as antimicrobial determinants by inhibiting the replication of fungi, bacteria, and viruses, without affecting the physiology of the host [21,22,23]. Furthermore, the use of plant extracts allows for (i) the mitigation of side effects and the improvement of patient tolerance [24], (ii) reduced production costs [25], (iii) the use of renewable compounds [26], and (iv) the assurance of a high degree of safety compared to synthetic drugs, among other benefits [27]. 

The trends in phytochemistry revealed that the Moraceae family showed multiple effects, as well as several other plants [28]. Among this family, the *Ficus* genus could be used for its antimicrobial, antidiabetic, anti-inflammatory, and analgesic activities, and also as a neuroprotective agent and antioxidant [29,30]. Furthermore, in combination with the health benefits, the absence of reported serious adverse events is an additional benefit [30].

The antimicrobial properties of several *Ficus* spp. have been extensively studied. The ethanol extract of *Ficus religiosa* L. (*F. religiosa*) leaves was active against *Bacillus subtilis*, *Staphylococcus aureus*, *Escherichia coli (E. coli)*, *Pseudomonas aeruginosa* (*P. aeruginosa*), and *Salmonella typhimurium* [31,32]. *F. carica* L. green fruit latex has been examined by Aref et al. for its activities against pathogenic yeasts. The methanol fraction inhibited the growth of *Candida albicans, Cryptococcus neoformans, and Microsporum canis* [33]. Regarding antiviral properties, Yarmolinsky et al. showed the antiviral activity of *F. benjamina* L. against HSV-1, HSV-2, and Varicella-Zoster Virus (VZV) when the extracts were added to the cells at the time of infection [34]. Moreover, *Ficus pumila* (*F. pumila*) L. extract was effective against human T-cell leukemia virus type 1. Patients orally administered with *F. pumila* extract showed improved symptoms and prognoses compared to untreated ones [35].

Port Jackson fig leaf extract (*Ficus rubiginosa* Desf. ex Vent.) exhibited antimicrobial activities against *P. aeruginosa* [36] and an intermediate growth inhibition of *C. albicans* [37]. Furthermore, a slight susceptibility to *F. rubiginosa* ethanolic extract was observed against *Aeromonas hydrophila* [38,39].

Until now, no evidence concerning the antiviral properties of *F. rubiginosa* plant extract has been reported. Therefore, in the present study, the antiviral activity of the methanolic extract (M) and its *n*-hexane (H) and ethyl acetate (EA) fractions against DNA/RNA and with/without envelope viral strains was studied.

## 2. Materials and Methods

### 2.1. Plant Material

Fresh leaves of *Ficus rubiginosa* Desf. ex Vent. were collected at the Botanical Garden of Padua on June 2018 by Prof. Piovan and immediately lyophilized (Virtis Benchtop BT 2K ES). A voucher specimen (H0061262) was deposited at the Botanical Garden of Padua. The leaves were pulverized in a mechanical grinder (TR2000, Girmi)*,* and the resulting powder was passed through a 710 µm wire sieve.

### 2.2. Extraction and Solvent Partition

The powdered leaves (56.5 g) were extracted by maceration with MeOH (3 × 570 mL) overnight, at room temperature, yielding 4.6 g (8.13% yield) of methanolic extract (M). M was dissolved with MeOH (50 mL), diluted with H_2_O (60 mL), and sequentially extracted with *n*-hexane (4 × 50 mL) and ethyl acetate (4 × 50 mL). All the organic extracts were dried over anhydrous Na_2_SO_4_ (Carlo Erba, Milan, Italy), filtered, and evaporated under reduced pressure yielding 1.27 g (H) and 0.11 g (EA). All the solvents were purchased from VWR (Milan, Italy).

### 2.3. Cell Line and Viral Strains

The cell line derived from the renal epithelium of the African green monkey (*Cercopithecus aethiops*) (VERO CCL-81) was purchased from the American Type Culture Collection (ATCC). Cells were cultured in Dulbecco’s Modified Eagle Medium (DMEM; Gibco; Thermo Fisher Scientific, Waltham, MA, USA) with 4.5 g/L of glucose, 2 mM of L-glutamine, 100 IU/mL of penicillin-streptomycin solution, and supplemented with 10% fetal bovine serum (FBS; Gibco; Thermo Fisher Scientific, Waltham, MA, USA) in a humidified atmosphere with 5% CO_2_ at 37 °C.

The viruses used were HSV-1 SC16 containing a lacZ gene driven by the cytomegalovirus IE-1 promoter to express beta-galactosidase, HSV-1 containing the Green Fluorescent Protein reporter inserted into the gene encoding the integument protein VP22 (HSV-1-GFP), HCoV-229E (ATCC VR-740), and PV-1 (Table 1). All viruses were propagated in the VERO CCL-81 monolayer.

### 2.4. Cytotoxic Activity

The cytotoxicity of the VERO CCL-81 cells was determined using the 3-(4,5-dimethylthiazol-2-yl)-2,5-diphenyl tetrazolium bromide (MTT) assay. Cells were seeded at a density of 2 × 10^4^ cells/well into 96-well microtiter plates and incubated at 37 °C with 5% CO_2_ in a humid environment. The following day, cells were treated with M (1400–10 µg/mL), H (600–10 µg/mL), and EA (100–10 µg/mL) for 24, 48, and 72 h. The cells treated with the solvent used to dissolve the plant material represented the negative control (CTRL−), while the cells treated with DMSO (100%) constituted the positive control (CTRL+). After incubation, 100 µL of MTT solution (Sigma-Aldrich, St. Louis, MO, USA) (0.3 mg/mL) was added to each well for 3 h at 37 °C. The solution was removed, and 100 µL of DMSO (100%) was added to solubilize the formazan crystals. The absorbance at 570 nm was measured using a microplate reader (Tecan, Männedorf, Swiss), and the cytotoxicity percentage was calculated according to the following formula:$$\%\mathrm{Cytotoxicity}=100-[100\;\times \;(\frac{OD570\;\mathrm{nm}\;of\;the\;test\;sample}{OD570\;\mathrm{nm}\;of\;CTR-})]$$

### 2.5. Antiviral Activity

Four assays were performed to evaluate the antiviral effects of M, H, and EA: (i) virus pre-treatment, (ii) cells pre-treatment, (iii) co-treatment, and (iv) post-treatment. For all tests, VERO CCL-81 cells were plated at a density of 2 × 10^5^ cells/well in a 12-well plate and incubated overnight (ON) at 37 °C. The following day, the cells were treated with M, H, and EA in a concentration range between 10–0.08 µg/mL, while the viral strains were used at the infection multiplicity (MOI) of 0.01. The cells treated with the solvent used to dissolve the compounds represented the negative control (CTRL−), while several compounds were used as positive controls (CTRL+). Specifically, for HSV-1, melittin (5 µM) was used in the co-treatment and virus pre-treatment, dextran-sulfate (1 µM) in cell pre-treatment, and aciclovir (5 µM) in post-treatment; against HCoV-229E, rhamnolipids M15RL (50 µg/mL) were used in co-treatment and virus pre-treatment, ivermectin (10 µM) in cell pre-treatment, and remdesivir (10 µM) in post-treatment; for PV-1, pleconaril (2 µg/mL) was used in co-treatment and virus pre-treatment, and WIN51711 (5 µg/mL) and protein 2C (10 µM) in cell pre-treatment and post-treatment, respectively [40,41]. The activity of M, H, and EA against HSV-1, HCoV-229E, and PV-1 was evaluated by plaque reduction assays in infected cells.

(i)Virus pre-treatment assay: each compound was diluted in 1X phosphate-buffered saline (1X PBS) (Sigma-Aldrich, St. Louis, MO, USA) and pre-incubated with the viral suspension at 10^4^ Plaque-Forming Units (PFU) in DMEM without (w/o) FBS for 1 h at 37 °C. Subsequently, the mixture was diluted 1:10 in DMEM w/o FBS and used to infect the cell monolayer for 1 h at 37 °C.(ii)Cell pre-treatment assay: cells were pre-incubated in DMEM w/o FBS with each extract for 1 h at 37 °C. Then, the cells were infected with the viral suspension at 10^3^ PFU in DMEM w/o FBS for 1 h at 37 °C.(iii)Co-treatment assay: M, H, and EA, diluted to the selected concentrations, were co-incubated with the viral suspension at 10^3^ PFU directly on the cell monolayer in DMEM w/o FBS for 1 h at 37 °C.(iv)Post-treatment assay: the cells were previously infected with the viral suspension at 10^3^ PFU in DMEM w/o FBS for 1 h at 37 °C. Then, the cell monolayer was washed in 1X PBS to remove extracellular virions and treated with each extract for 1 h at 37 °C.

After adsorption, for each treatment, the cells were washed with citrate buffer (pH 3) (Sigma-Aldrich, St. Louis, MO, USA), and the culture medium supplemented with 5% carboxymethylcellulose (CMC) (Sigma-Aldrich, St. Louis, MO, USA) was added. Infected cells were incubated for 24 h for HCoV-229E and 48 h for HSV-1 and PV-1. The plaque count was performed by fixing the cells with 4% formaldehyde (Sigma-Aldrich, St. Louis, MO, USA) and then staining them with 0.5% crystal violet (Sigma-Aldrich, St. Louis, MO, USA). The plates were examined by plaque counting, and the percentage of viral inhibition was calculated according to the following formula:$$\%\mathrm{Viral}\mathrm{inhibition}=100-[100\;\times \;(\frac{plaques\;counted\;in\;the\;test\;sample}{plaques\;counted\;in\;the\;CTRL-})]$$

### 2.6. Gene Expression Analysis

To confirm the antiviral efficacy of M, H, and EA, a quantitative polymerase chain reaction (qPCR) was performed. The tests were conducted as described above. After incubation, the collected cells were subjected to total RNA extraction with TRIzol (Thermo Fisher, Waltham, MA, USA). The obtained RNA was quantified using the nanodrop (NanoDrop 2000, Thermo Fisher Scientific, Waltham, MA, USA) and 1 microgram was retrotranscribed in cDNA, according to the instructions of the SensiFAST ™ cDNA Synthesis Kit (Meridian Bioscience, Washington, DC, USA). The qPCR was performed in a total volume of 20 µL containing 0.3 µM of each primer, 1X BrightGreen qPCR MasterMix (abm, San Francisco, CA, USA), and 100 ng of cDNA. Amplification was conducted in Thermal Cycler UNO96 (VWR International, Radnor, PA, USA) using the following amplification program: denaturation at 95 °C for 15 s, annealing at 60 °C for 20 s, and extension at 72 °C for 15 s (40 cycles). The expressions of UL27 and UL54 (for HSV-1) and protein S and N (for HCoV-229E) were evaluated using the primer sequences reported in Table 2. The target threshold cycle (Ct) values were normalized to glyceraldehyde 3-phosphate dehydrogenase (GAPDH), used as a housekeeping-gene control. Gene expression was examined by calculating 2^−ΔΔCt^.

### 2.7. Statistical Analysis

All assays were conducted in triplicate and expressed as mean ± Standard Deviation (SD). The ordinary one-way ANOVA, Dunnett’s multiple comparison test, inhibition concentration 50% and 90% (IC_50_–IC_90_), and cytotoxic concentration 50% (CC_50_) were performed using GraphPad Prism ver. 8.2.1 for macOS (Software GraphPad, San Diego, CA, USA, www.graphpad.com, accessed on 2 September 2022). Dunnett’s multiple comparisons test expressed the significance of the differences between the samples treated versus the untreated sample (CTRL− vs. samples). Values were considered significant when *p*-value < 0.05.

## 3. Results

### 3.1. Cytotoxicity and Safety of F. rubiginosa Leaves Extract

The cytotoxic effects were evaluated in the VERO CCL-81 cell line after 24, 48, and 72 h of treatment with M (1400–10 µg/mL), H (600–10 µg/mL), and EA (100–10 µg/mL). The cytotoxic potential increased in a dose-dependent manner with a cell death rate greater than 50% at concentrations of 350, 300, and 50 µg/mL for M, H, and EA, respectively, after 72 h of treatment (Figure 1A–C). In detail, the recorded CC_50_ was 513.1, 298.6, and 56.45 µg/mL for M, H, and EA, respectively. The antiviral activity was evaluated at a concentration that did not affect cell viability; therefore, it was lower than 10 µg/mL.

### 3.2. Antiviral Activity by Plaque Reduction Assays

The antiviral activities of M, H, and EA from *F. rubiginosa* extract were investigated against HSV-1 (enveloped DNA virus), HCoV-229E (enveloped RNA virus), and PV-1 (RNA virus w/o envelope) by plaque reduction assays in infected VERO CLL-81 cells.

#### 3.2.1. Herpesviridae: HSV-1

In the co-treatment assay, each sample (M, H, and EA) was co-incubated with the virus directly on the cell monolayer. All compounds were effective against HSV-1, recording inhibition rates of 53%, 65%, and 47% at 0.31 µg/mL for M, H, and EA, respectively (Figure 2A–C). To better understand the viral target, virus- and cell-pre-treatment assays were performed. A strong inhibitory effect was observed by pre-incubating the virus with M, H, and EA, showing a complete inactivation up to 0.62, 0.31, and 1.25 µg/mL, respectively (Figure 2D–F). On the other hand, no inhibition was observed by pre-incubating VERO CCL-81 cells with each sample, which were subsequently infected (Figure 2G–I). These results indicated that the extracts could be effective in the extracellular infection phase, altering the viral structure. To confirm this hypothesis and exclude an intracellular action, a post-treatment assay was conducted. In the infected VERO CCL-81 cells, none of the extracts were capable of inhibiting viral replication (Figure 2J–L). In Table 3 (A), the IC_50_ and IC_90_ values relating to each test performed are reported. The plaque reduction assay was confirmed by fluorescence microscopy (Figure 3). In detail, an additional virus pre-treatment assay was performed using HSV-1 engineered with GFP, which dyes the infected cells a fluorescent green. The results were shown for each treatment through bright-field (RGB) and fluorescent images in a concentration range of 0.62–0.08 µg/mL. From the captured images, it was evident that at the concentration of 0.62 µg/mL, M and H showed a strong inhibitory effect, and, therefore, no fluorescence signal was recorded; conversely, the EA fraction, which had an activity of 55%, showed a slight fluorescence signal. The cytopathic effect gradually increased in the concentration range of 0.31–0.16 µg/mL. Precisely at the concentration of 0.08 µg/mL, the high fluorescence signal was comparable to the virus control. These results were consistent with the data obtained in the plaque reduction test.

#### 3.2.2. Coronaviridae: HCoV-229E

To investigate the role of M, H, and EA against enveloped RNA viruses, the degree of antiviral activity was tested using a viral model belonging to the *Coronaviridae* family. For HCoV-229E, the same experimental conditions previously described were conducted. Similar results to those for HSV-1 were recorded, but with less antiviral efficacy. In detail, when M, H, and EA were incubated simultaneously with the virus on the cell monolayer, levels of viral inhibition of 48, 55, and 40% were recorded, respectively, at the concentration of 2.5 µg/mL (Figure 4A–C). By pre-treating the virus for 1 h with each sample, improved activity was observed, with a total inhibition of replication up to the concentration of 1.25 µg/mL for M and H and 5 µg/mL for EA (Figure 4D–F). Similar to HSV-1, no activity was recorded in the cell pre-treatment and post-treatment assays, indicating that no fraction interacted with the cell membrane and interfered with HCoV-229E’s viral replication phases (Figure 4G–L). In Table 3 (B), the IC_50_ and IC_90_ values corresponding to each test performed are reported.

#### 3.2.3. Picornaviridae: PV-1

The results obtained against HSV-1 and HCoV-229E in the virus co-treatment and virus pre-treatment assays suggested that M, H, and EA’s action occurred in the extracellular viral infection phase, interacting directly with the viruses’ structures. To better understand the target of these compounds, PV-1 was chosen as a non-enveloped RNA virus. Antiviral activity against PV-1 was tested in the same experimental conditions used for HSV-1 and HCoV-229E. However, no relevant viral inhibition was recorded in the tested concentration range (10–0.08 µg/mL), indicating that M, H, and EA could act directly on the viral envelope, destructuring the lipid membrane (Table 4).

### 3.3. Analysis of Viral Gene Expression

To confirm the results obtained from the plaque reduction assay, qPCR was performed. Regarding HSV-1, the UL54 and UL27 genes were investigated. The first is an immediate gene encoding the ICP27 protein, which inhibits mRNA splicing and promotes the nuclear export of viral transcripts. UL27 is a late gene encoding the structural glycoprotein B (gB). For the molecular analysis, the virus pre-treatment assay was performed under the experimental conditions described for the plaque reduction assay. After 30 h, the RNA was extracted, and the cDNA was synthesized. The real-time PCR showed that the expression of both genes was inhibited up to 1.25 µg/mL for M and H and up to 2.5 µg/mL for EA (Figure 5). Furthermore, the gene expression increased in a dose-dependent manner, reaching the same value as the virus control at 0.31 µg/mL for M and EA. Regarding H, at the concentration of 0.31 µg/mL an expression of 50% of UL27 and UL54 was recorded. Overall, the verified gene inhibition suggested that the treatment with M, H, and EA impaired HSV-1 replication. Then, the activity of each sample was confirmed against HCoV-229E by analyzing the expression levels of the genes encoding the proteins S and N, which are involved in the virus’ entry into the cell and in the packaging of the viral genome, respectively. The real-time PCR, in accordance with the results obtained from the plaque assays, showed that the infection was inhibited in a dose-dependent manner. In detail, a complete inhibition up to 2.5, 1.25, and 5 µg/mL was observed for M, H, and EA, respectively, with a linear increase at subsequent concentrations (Figure 6).

## 4. Discussion

In this study, we investigated in vitro the antiviral potential of *F. rubiginosa* leaf extract using three different viral models belonging to the *Herpesviridae*, *Coronaviridae,* and *Picornaviridae* families.

The *F. rubiginosa* leaf samples were tested against HSV-1, HCoV-229E, and PV-1 at different stages of viral infection through virus pre-treatment, co-treatment, cell pre-treatment, and post-infection assays.

Plant extracts are believed to exert their inhibitory action early in the viral infection cycle, mainly during virus absorption and/or host cell penetration [42,43]. In accordance, our results demonstrated that the *F. rubiginosa* samples were active against HSV-1 and HCoV-229E through direct interaction with viral particles and by blocking the viruses’ access to the host cells. This action mode was similar for both enveloped viruses but not for naked viruses, such as PV-1, suggesting that M, H, and EA could act on the outermost viral structure, particularly on the envelope glycoproteins. The fractionation of the extract improved its antiviral effectiveness compared to the raw extract (M). A superior inhibitory effect was obtained with H. In detail, a complete viral inhibition was recorded up to concentrations of 0.31 and 1.25 µg/mL against HSV-1 and HCoV-229E, respectively, in the virus pre-treatment assay. On the other hand, the EA fraction was active up to 1.25 and 5 µg/mL against HSV-1 and HCoV-229, respectively. These results suggested that H could represent the richest bioactive compound fraction, inhibiting both enveloped viral strains. Regarding the latter, a powerful sample inhibition against HSV-1 compared to HCoV-229E was observed. Indeed, in the pre-virus treatment, the M, H, and EA antiviral efficiencies against HSV-1 were greater compared to HCoV-229E. The explanation could be associated with the different structural envelope compositions of the two viruses. For example, HSV-1 contains 15 viral proteins in its lipid envelope, of which 12 are glycosylated and 3 non-glycosylated. Four of the glycosylated proteins—gD, gH, gL, and gB—are essential for entry into target cells [44]. HCoV-229E, HCoV-NL63, and SARS-CoV exhibit four structural proteins (S, M, N, and E) [45]. The *F. rubiginosa* samples’ affinity towards membrane glycoproteins and/or the different lipid envelopes could lead to changes in efficiency in the two viral models. To date, there are still no studies that have investigated the role of *F. rubiginosa* extracts as antiviral agents; however, such research appears essential, particularly in light of the antiviral effect found with other *Ficus* species extracts. Yarmolinsky et. al. reported that raw ethanol extracts from *F. benjamina* strongly inhibited cellular HSV-1 and -2 and Varicella Zoster Virus (VZV) infections in vitro when cells were treated during and after infection [34].

The anti-HSV-1 effect was observed in vitro using the water extract from *F. carica*, which showed low toxicity and a direct virus-killing effect [46]. Other extracts of *F. carica* were tested against HSV-1, echovirus type 11, and adenovirus. At the concentrations of 78 µg/mL, the H and EA fractions inhibited the multiplication of such viruses, showing no cytotoxic effect on the Vero cells [47].

Another in vitro study demonstrated that aqueous and chloroform bark extracts of *F. religiosa* were active against HSV-2 and acyclovir-resistant strains. The results related to the chloroform extract showed a direct inactivation of viral activity, hindering the virus’ entry into the host cell and inhibiting viral proliferation [48]. Different results were shown in the Vero cells incorporating the MeOH extract of the aerial parts of *F. vasta,* which was not active against HSV-1 or bacteria such as *S. aureus*, *S. epidermidis*, *E. coli,* and *P. aeruginosa* [49]. An in vitro study investigated the flavonoid-rich extracts isolated from *Ficus ischnopoda* L. and demonstrated that some of these exhibited a significant anti-HSV-1 activity [50].

Despite the constant development of effective treatments, numerous problems still exist concerning fighting viral infections. Moreover, the high costs, the drug safety, the side effects, and the onset of drug resistance require the development of new alternative strategies [51] to already approved drugs [52]. Our results highlight that the *Ficus* genus could represent an interesting natural resource of antiviral compounds. A limitation of this study is that a phytochemical analysis of the extracts was not performed and the mechanism of action underlying the demonstrated viral activity was not investigated. Further investigations will be needed to better understand the active molecules responsible for antiviral activity and their specific mechanism of action against HSV-1 and HCoV-229E.

## Figures and Tables

**Figure 1 viruses-14-02257-f001:**
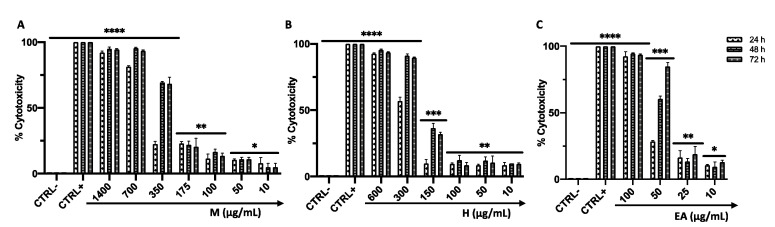
Cytotoxic effects (%) of M (**A**), H (**B**), and EA (**C**) in VERO CCL-81 cells after 24, 48, and 72 h of treatment. The data represent the mean ± standard deviation (SD) of three independent experiments. ****: *p*-value < 0.0001; ***: *p*-value < 0.0003; **: *p*-value < 0.008; *: *p*-value < 0.03.

**Figure 2 viruses-14-02257-f002:**
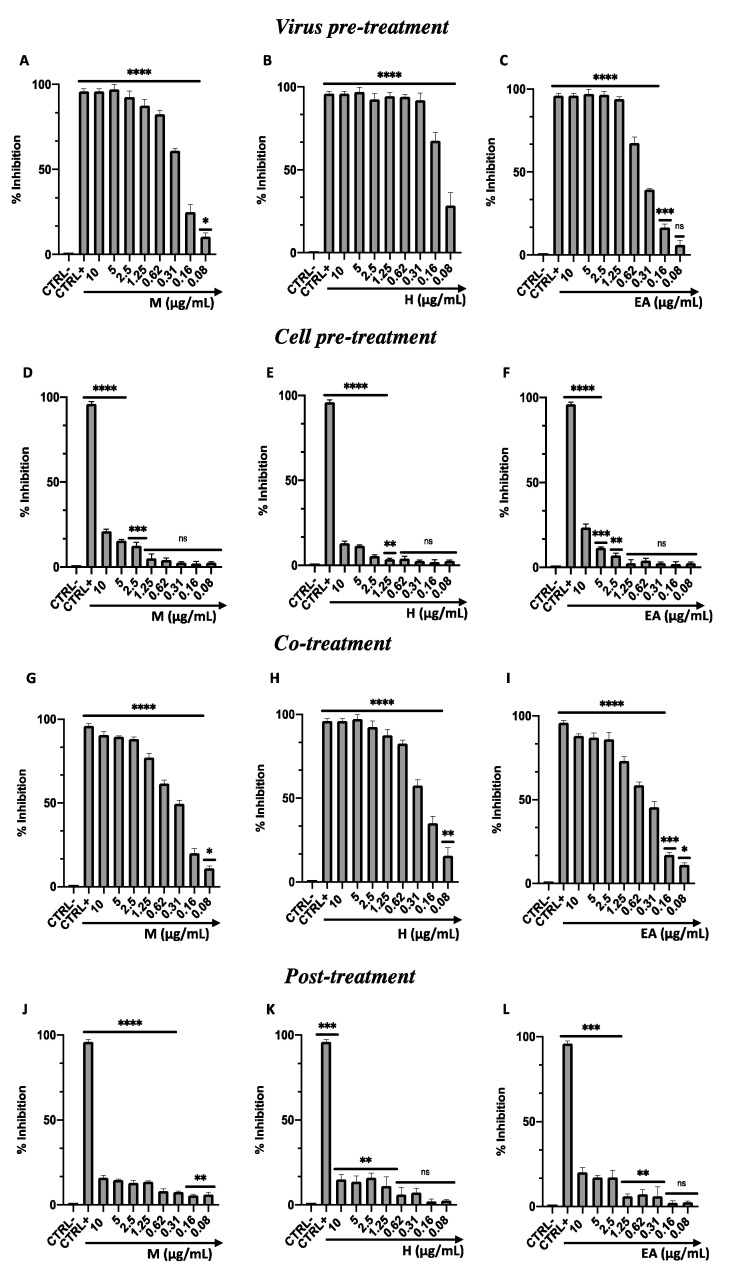
Antiviral activity of M, H, and EA from *F. rubiginosa* L. extract against HSV-1. Four plaque reduction assays were performed: (**A**–**C**) Virus pretreatment assay; (**D**–**F**) Cell pretreatment assay; (**G**–**I**) Co-treatment assay; (**J**–**L**) Post-treatment assay. Data represent mean ± standard deviation (SD) of three independent experiments. ****: *p*-value < 0.0001; ***: *p*-value < 0.0009; **: *p*-value < 0.006; *: *p*-value < 0.04; ns: not significant.

**Figure 3 viruses-14-02257-f003:**
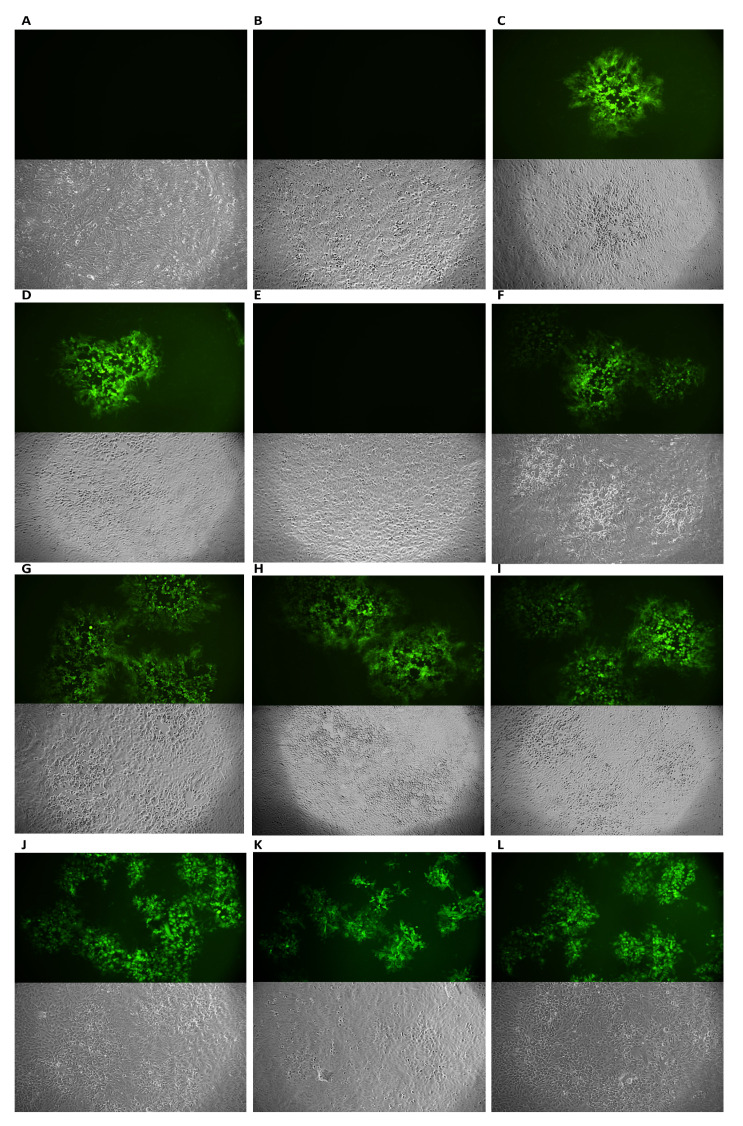
Antiviral activity of M, H, and EA, against HSV-1-GFP. Plaques were viewed with RGB and fluorescent microscopy. (**A**–**C**): M, H, and EA treatments at a 0.62 µg/mL; (**D**–**F**): M, H, and EA treatments at a 0.31 µg/mL; (**G**–**I**): M, H, and EA treatments at a 0.16 µg/mL; (**J**–**L**): M, H, and EA treatments at a 0.08 µg/mL; CTRL−: cells infected with the virus; CTRL+: cells uninfected.

**Figure 4 viruses-14-02257-f004:**
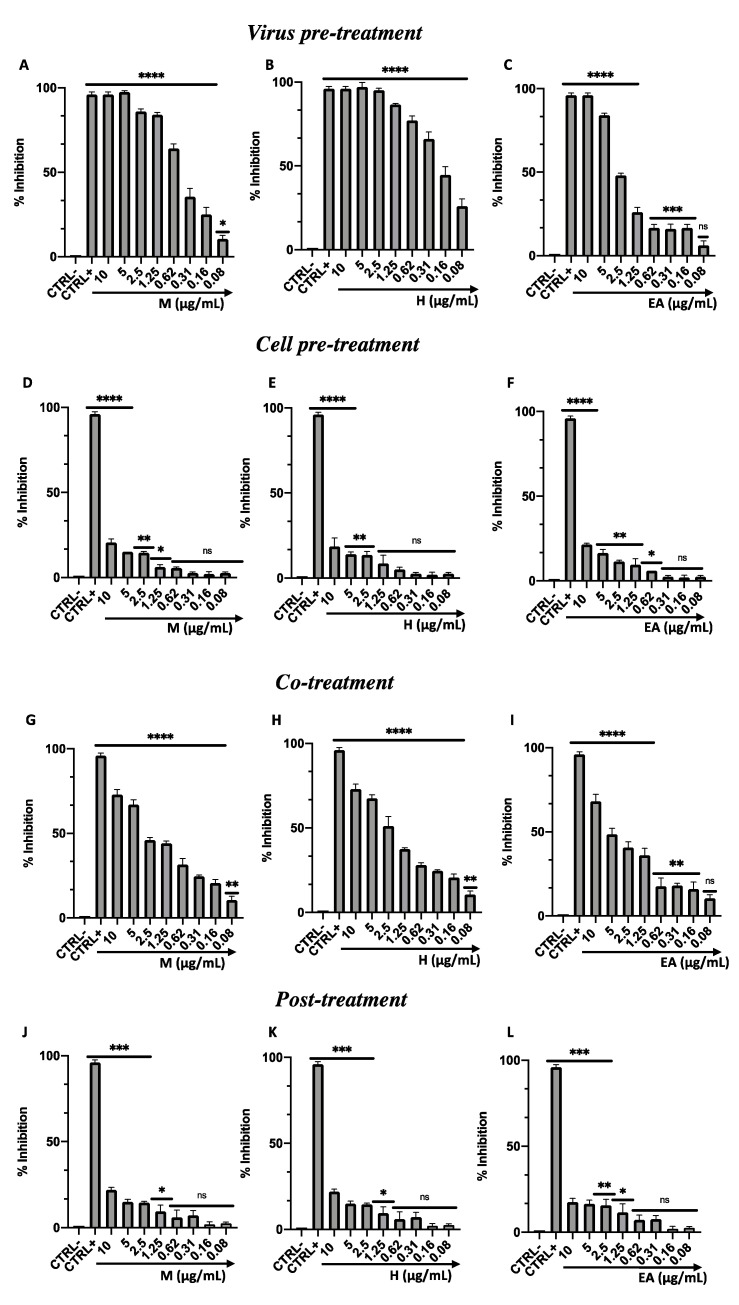
Antiviral activity of M, H, and EA from *F. rubiginosa* L. extract against HCoV-229E. Four plaque reduction assays were performed: (**A**–**C**) virus pretreatment assay; (**D**–**F**) cell pretreatment assay; (**G**–**I**) co-treatment assay; (**J**–**L**) post-treatment assay. Data represent mean ± standard deviation (SD) of three independent experiments. ****: *p*-value < 0.0001; ***: *p*-value < 0.0007; **: *p*-value < 0.008; *: *p*-value < 0.04; ns: not significant.

**Figure 5 viruses-14-02257-f005:**
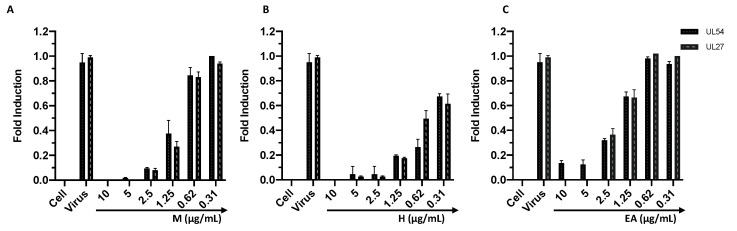
Analysis of HSV-1 UL27 and UL54 expression levels after treatment with M (**A**), H (**B**), and EA (**C**). Data represent the mean ± standard deviation (SD) of three independent experiments.

**Figure 6 viruses-14-02257-f006:**
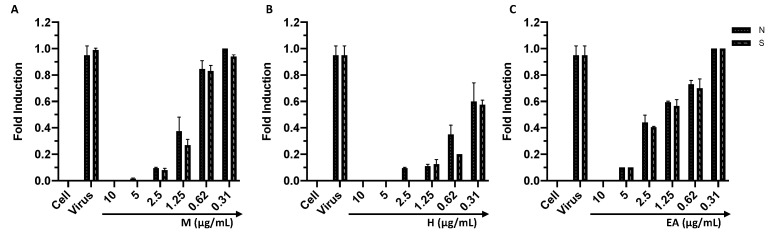
Analysis of HCoV-229E S and N expression levels after treatment with M (**A**), H (**B**), and EA (**C**). Data represent the mean ± standard deviation (SD) of three independent experiments.

**Table 1 viruses-14-02257-t001:** Main characteristics of the viral species used to evaluate the antiviral activity.

Viral Strain	Family	Nucleic Acid	Symmetry	Envelope	Dimension
HSV-1	*Herpesviridae*	dsDNA	icosahedral	yes	115–240 nm
HCoV-229E	*Coronaviridae*	ssRNA (+)	helical	yes	80–120 nm
PV-1	*Picornaviridae*	ssRNA (+)	icosahedral	no	30–40 nm

**Table 2 viruses-14-02257-t002:** Primer sequences of HSV-1 and HCoV-229E used for gene expression analysis.

Virus	Gene	Forward Sequence	Reverse Sequence
HSV-1	UL54	5′-TGGCGGACATTAAGGACATTG-3′	3′-TGGCCGTCAACTCGCAG-5′
HSV-1	UL27	5′-GCCTTCTTCGCCTTTCGC-3′	3′-CGCTCGTGCCCTTCTTCTT-5′
HCoV-229E	S	5′-CGTTGAACTTCAAACCTCAGA-3′	3′-ACCAACATTGGCATAAACAG-5′
HCoV-229E	N	5′-GTCGTCAGGGGTAGAATACCTTA-3′	3′-CCCGTTTGCCCTTTCTAGT-5′
	GAPDH	5′-CCTTTCATTGAGCTCCAT-3’	3′-CGTACATGGGAGCGTC-5’

**Table 3 viruses-14-02257-t003:** IC_50_ and IC_90_ of M, H, and EA against HSV-1 (A) and HCoV-229E (B) in virus pre-treatment, cell pre-treatment, co-treatment, and post-treatment assays.

A	Virus Pre-Treatment HSV-1					B	Virus Pre-Treatment HCoV-229E				
M (µg/mL)		H (µg/mL)		EA (µg/mL)		M (µg/mL)		H (µg/mL)		EA (µg/mL)	
IC_50_	IC_90_	IC_50_	IC_90_	IC_50_	IC_90_	IC_50_	IC_90_	IC_50_	IC_90_	IC_50_	IC_90_
0.33	0.73	0.125	0.24	0.455	0.89	0.50	1.14	0.20	0.78	2.60	5.52
**Cell Pre-treatment HSV-1**						**Cell Pre-treatment HCoV-229E**					
**M (µg/mL)**		**H (µg/mL)**		**EA (µg/mL)**		**M (µg/mL)**		**M (µg/mL)**		**M (µg/mL)**	
IC_50_	IC_90_	IC_50_	IC_90_	IC_50_	IC_90_	IC_50_	IC_50_	IC_50_	IC_50_	IC_50_	IC_50_
>10	>10	>10	>10	>10	>10	>10	>10	>10	>10	>10	>10
**Virus Co-treatment HSV-1**						**Cell Pre-treatment HCoV-229E**					
**M (µg/mL)**		**H (µg/mL)**		**EA (µg/mL)**		**M (µg/mL)**		**M (µg/mL)**		**M (µg/mL)**	
IC_50_	IC_90_	IC_50_	IC_90_	IC_50_	IC_90_	IC_50_	IC_50_	IC_50_	IC_50_	IC_50_	IC_50_
0.51	1.33	0.29	0.67	0.63	1.62	3.70	3.70	3.70	3.70	3.70	3.70
**Post-treatment HSV-1**						**Post-treatment HCoV-229E**					
**M (µg/mL)**		**H (µg/mL)**		**EA (µg/mL)**		**M (µg/mL)**		**H (µg/mL)**		**EA (µg/mL)**	
IC_50_	IC_90_	IC_50_	IC_90_	IC_50_	IC_90_	IC_50_	IC_90_	IC_50_	IC_90_	IC_50_	IC_90_
>10	>10	>10	>10	>10	>10	>10	>10	>10	>10	>10	>10

**Table 4 viruses-14-02257-t004:** Antiviral activity of M, H, and EA from *F. rubiginosa* leaves extract against PV-1.

Virus Pre-Treatment PV-1 (% Inhibition)		
M (10 µg/mL)	H (10 µg/mL)	EA (10 µg/mL)
12%	15%	9%
**Co-treatment PV-1 (% Inhibition)**		
**M (10 µg/mL)**	**H (10 µg/mL)**	**EA (10 µg/mL)**
9%	15%	11%
**Cell pre-treatment PV-1 (% Inhibition)**		
**M (10 µg/mL)**	**H (10 µg/mL)**	**EA (10 µg/mL)**
5%	6%	5%
**Post-treatment PV-1 (% Inhibition)**		
**M (10 µg/mL)**	**H (10 µg/mL)**	**EA (10 µg/mL)**
9%	7%	5%

## Data Availability

Not applicable.

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
