# Peer review of "Antiviral Activity of Ficus rubiginosa Leaf Extracts against HSV-1, HCoV-229E and PV-1"

_viruses, 2022, doi:10.3390/v14102257_

Round 1
Reviewer 1 Report
The manuscript „ Antiviral Activity of Ficus rubiginosa Leaf Extracts against HSV-1 and HCoV-229E” is of great importance, the need for new antiviral drugs with specific effects is modern. Plant-based antiviral drugs may be a potential source of drug development, so the authors conducted the relevant research in line with this assumption. Currently, there is a lack of research on the antiviral properties of the plant extract of F. rubiginosa. Therefore, the authors of the manuscript conducted a study of the antiviral activity of the methanolic extract of this plant and its n-hexane and ethyl acetate fractions against DNA and RNA of both enveloped and non-enveloped viruses.
The strength of the presented research is to show that the genus Ficus can be a natural resource of antiviral compounds against HSV-1 and HCoV-229E and to indicate the need for further research to understand the active compounds and the mechanism of their action.
The manuscript is clear, relevant to the field, and presented in a well-structured manner. Most of the citations come from articles published in the last 5 years. The manuscript is scientifically substantiated, and the design of the experiment is suitable for testing the hypothesis. The conclusions are in line with the evidence presented. The formulated statements and conclusions are consistent and supported by the quotes mentioned.
I have a few comments to the authors suggesting changes to the manuscript:
· Two viruses are mentioned in the manuscript title and keywords: HSV-1 and HCoV-229E, while research was also carried out for PV-1. Also in the introduction, there is no information about the PV-1.
· In lines 83 and 85 [31] it is repeated.
· In lines: 137, 138, 153 - there is "CTR" and in lines: 152, 210, 263 and in Figure 3 it is: "CTRL" - please standardize it.
· The text fragments in lines 85 and 380 are a repetition of the quotation.
· Line 217 and 218 - please check 350 µg/mL and 513.1 µg/mL, are they consistent with Figure 1A?
· Line 36: "with antiviral efficiency against HSV-1 four times greater compared to HCoV-229E" - what does this conclusion result from?
· Line 368: “the M, H, and EA antiviral efficiency against HSV-1 were four times greater compared to HCoV-229E, in pre-virus treatment” - why this conclusion?
· I suggest Table 3 be placed in the manuscript before Figure 2.
· I also propose that the sequence of experiments described in paragraph “2.5. Antiviral activity ”, in Tables 3 and 4 and Figures 2 and 4 was the same, e.g .: i) Virus pre-treatment assay, ii) Cell pre-treatment assay, iii) Co-treatment assay, iv) Post-treatment assay or as in tables and figures.
· In Table 3 values > 50 µg/mL are given for the IC50 and IC90 while the antiviral activity was tested in the range of 10-0.08 µg/mL, so why these data? There is also no comment on the obtained IC90 value in the description of the study results.
· I suggest that Tables 3 and 4 be in a more readable form or that the data from Table 4 be presented in the form of graphs.
Author Response
Reviewer 1
The manuscript “Antiviral Activity of Ficus rubiginosa Leaf Extracts against HSV-1 and HCoV-229E” is of great importance, the need for new antiviral drugs with specific effects is modern. Plant-based antiviral drugs may be a potential source of drug development, so the authors conducted the relevant research in line with this assumption. Currently, there is a lack of research on the antiviral properties of the plant extract of F. rubiginosa. Therefore, the authors of the manuscript conducted a study of the antiviral activity of the methanolic extract of this plant and its n-hexane and ethyl acetate fractions against DNA and RNA of both enveloped and non-enveloped viruses. The strength of the presented research is to show that the genus Ficus can be a natural resource of antiviral compounds against HSV-1 and HCoV-229E and to indicate the need for further research to understand the active compounds and the mechanism of their action. The manuscript is clear, relevant to the field, and presented in a well-structured manner. Most of the citations come from articles published in the last 5 years. The manuscript is scientifically substantiated, and the design of the experiment is suitable for testing the hypothesis. The conclusions are in line with the evidence presented. The formulated statements and conclusions are consistent and supported by the quotes mentioned.
I have a few comments to the authors suggesting changes to the manuscript:
1.R: Two viruses are mentioned in the manuscript title and keywords: HSV-1 and HCoV-229E, while research was also carried out for PV-1. Also in the introduction, there is no information about the PV
1.A: We thank the reviewer for the advice, we have modified the Title, keywords, abstract (line 37) and introduction paragraph (lines 79-83) as suggested.
2.R: In lines 83 and 85 [31] it is repeated
2.A: We thank the reviewer for his careful observation. We have removed the repeated reference 31
3.R: In lines: 137, 138, 153 - there is "CTR" and in lines: 152, 210, 263 and in Figure 3 it is: "CTRL" - please standardize it
3.A: We apologize for the error. We have standardized “CTRL”.
4.R: The text fragments in lines 85 and 380 are a repetition of the quotation
4.A: We are grateful to the reviewer for the correction. We have modified the sentences in the manuscript (lines 90-92, lines 394-396).
5.R: Line 217 and 218 - please check 350 μg/mL and 513.1 μg/mL, are they consistent with Figure 1A?
5.A: We thank the reviewer for the observation. We checked figure 1A and after 72h of exposure, the cytotoxicity rate was greater than 50% at 350 μg/mL. In addition, the CC50 calculated via Graphpad software was 513.1 μg/mL.
6.R: Line 36: "with antiviral efficiency against HSV-1 four times greater compared to HCoV-229E" - what does this conclusion result from? 7.R: Line 368: “the M, H, and EA antiviral efficiency against HSV-1 were four times greater compared to HCoV-229E, in pre-virus treatment” - why this conclusion?
6.A-7.A: We completely agree with the reviewer and we apologize for the error. In fact, the antiviral efficiency of M, H, EA against HSV-1 was about 2, 4, 40 times higher than the HCoV-229E. For this reason, we have changed the sentences to indicate greater overall efficiency against HSV-1 compared to HCoV-229E (line 40, line 415).
8.R: I suggest Table 3 be placed in the manuscript before Figure 2.
8.A: We thank the reviewer for the advice, however, we placed Table 3 after figure 2 for formatting reasons.
9.R: I also propose that the sequence of experiments described in paragraph “2.5. Antiviral activity ”, in Tables 3 and 4 and Figures 2 and 4 was the same, e.g .: i) Virus pre-treatment assay, ii) Cell pre-treatment assay, iii) Co-treatment assay, iv) Post-treatment assay or as in tables and figures.
9.A: We thank the reviewer for his careful observation. We modified Tables 3 and 4 and Figures 2 and 4 following the experimental order described in paragraph 2.5
10.R: In Table 3 values > 50 μg/mL are given for the IC50 and IC90 while the antiviral activity was tested in the range of 10-0.08 μg/mL, so why these data? There is also no comment on the obtained IC90 value in the description of the study results.
10.A: We thank the reviewer for a careful review of the manuscript. CC50 values were calculated with GraphPad software, following nonlinear regression (curve fit), and dose-response inhibition. Otherwise, the value of IC90 was calculated using the GraphPad (https://www.graphpad.com/quickcalcs/Ecanything1.cfm) online software. The concentration at which 50% inhibition occurred was greater than the maximum concentration tested (10 μg /mL), in cell pre-treatment and post-treatment. We have corrected the manuscript.
11.R: I suggest that Tables 3 and 4 be in a more readable form or that the data from Table 4 be presented in the form of graphs
11.A: We are grateful to the reviewer for the suggestion and completely agree. We have modified Table 3 in accordance with your previous suggestions. We also modified Table 4 specifying that the% inhibition of each compound in the 4 treatments against the poliovirus referred to the maximum concentration used (10 μg / mL).

Reviewer 2 Report
In this manuscript, three fractions (M, H and EA) from Ficus rubiginosa leaf were obtained and their antiviral activity against HSV-1, HCoV-229E and PV-1 was investigated in the different phases of viral infection in Vero cell line. The results showed these fractions have good antiviral activity against HSV-1 (enveloped DNA virus) and HCoV-229E (enveloped RNA virus) but not PV-1 (RNA virus without envelope), suggesting that the leaf extract could act directly on the viral envelope. Overall, it can be a valuable contribution to the field. However, several points require attention and should be addressed as described below.
1. In Figure 2 and 4, some positive controls were used for antiviral activity. Can the authors provide some information about their mode of action? However, there are no positive controls in Table 4. Are there some positive controls for PV-1?
2. From Table 3, the extract H has a better inhibitory effect than M and EA and extract could act directly on the viral envelope. A proper discussion on which active ingredients might work and how to isolate the active ingredients from H extract is helpful.
3. Three-line table is recommended to use. Please correct all tables in this manuscript.
4. For statistical analysis, the authors showed that ****: p-value < 0.0001; ***: p-value = 0.0002; ** p-value = 0.0082; * p-value = 0.0460. Why are they so defined?
5. Other errors.
Please correct "700.000" in line 49, "(E. Coli)" in line 82, "50 μg / mL" in line 155, "2μg / mL" in 157, "5 μg / mL" in line 158, and "HSV-II" in line 388.
Please use 1X PBS instead of PBS1X in line 162 and 174.
Please check "1X BrightGreen 2X qPCR MasterMix" in line 192.
Please add punctuation in line 220 and correct punctuation in line 241-243.
Author Response
Reviewer 2
In this manuscript, three fractions (M, H and EA) from Ficus rubiginosa leaf were obtained and their antiviral activity against HSV-1, HCoV-229E and PV-1 was investigated in the different phases of viral infection in Vero cell line. The results showed these fractions have good antiviral activity against HSV-1 (enveloped DNA virus) and HCoV-229E (enveloped RNA virus) but not PV-1 (RNA virus without envelope), suggesting that the leaf extract could act directly on the viral envelope. Overall, it can be a valuable contribution to the field. However, several points require attention and should be addressed as described below.
1 R: In Figure 2 and 4, some positive controls were used for antiviral activity. Can the authors provide some information about their mode of action? However, there are no positive controls in Table 4. Are there some positive controls for PV-1?
1 A: We thank the reviewer for the careful analysis of the experimental setup. The controls used for HSV-1, HCoV-229E and PV-1 in the different treatment phases are explained in the section "Materials and methods, paragraph 2.5". “In detail, for HSV-1 melittin (5 μM) in co-treatment and virus pre-treatment, dextran-sulfate (1 μM) in cell pre-treatment and aciclovir (5 μM) in post-treatment; Against HCoV-229E, rhamnolipids M15RL (50 μg / mL) in co-treatment and virus pre-treatment, ivermectin (10 μM) in cell pre-treatment and remdesivir (10 μM) in post-treatment; for PV-1, pleconaril (2μg / mL) in co-treatment and virus pre-treatment, WIN51711 (5 μg / mL) and protein 2C (10 μM) in cell pre-treatment and post-treatment, respectively.” They represent standardized controls reported in the literature, able to act in the different stages of infection. In materials and methods, I have added the bibliographic references associated with the controls used. Also, for PV-1 the controls used were reported in Materials and methods, in table 4 I have only explained the percentage of inhibition obtained at the highest concentration (10 μg/mL). I changed table 4 to make it clearer.
2.R: From Table 3, the extract H has a better inhibitory effect than M and EA and extract could act directly on the viral envelope. A proper discussion on which active ingredients might work and how to isolate the active ingredients from H extract is helpful.
2.A: We thank the reviewer for raising a very interesting issue. However, in this study, the phytochemical analysis of the extracts was not performed and the mechanism of action underlying the demonstrated viral activity was not investigated. This was added as a limitation at the end of the discussion paragraph. We would like to further investigate the answers to these questions in a future study that includes a complete analysis of the active compounds and all potential targets.
3.R: Three-line table is recommended to use. Please correct all tables in this manuscript.
3. A: We thank the reviewer for the suggestion, however we believe that the use of such a table may result in a loss of some useful data, moreover this type of table is not recommended in the submission instructions.
4. R: For statistical analysis, the authors showed that ****: p-value < 0.0001; ***: p-value = 0.0002; ** p-value = 0.0082; * p-value = 0.0460. Why are they so defined?
4. A: We agree with the reviewer. Previously we had performed an exact average of the p-values obtained for each treatment. Now, we have modified, defining the p-value ranges for each assay, considering the higher values.
5.R: Please correct "700.000" in line 49, "(E. Coli)" in line 82, "50 μg / mL" in line 155, "2μg / mL" in 157, "5 μg / mL" in line 158, and "HSV-II" in line 388.
6: R: Please use 1X PBS instead of PBS1X in line 162 and 174.
7:R: Please check "1X BrightGreen 2X qPCR MasterMix" in line 192.
8.R: Please add punctuation in line 220 and correct punctuation in line 241-243.
5-8 A: We thank the reviewer for careful corrections. We have edited errors in the manuscript as required.
